# Measuring the Effectiveness of a Multicomponent Program to Manage Academic Stress through a Resilience to Stress Index

**DOI:** 10.3390/s23052650

**Published:** 2023-02-28

**Authors:** Carlos Figueroa, Andrés Ayala, Luis A. Trejo, Bertha Ramos, Clara L. Briz, Isabella Noriega, Alejandro Chávez

**Affiliations:** 1School of Health, Tecnologico de Monterrey, Ciudad de México 14380, Mexico; 2Vicerrectoría de Innovación Educativa y Normatividad Académica, Tecnologico de Monterrey, Monterrey 64849, Mexico; 3School of Engineering and Sciences, Tecnologico de Monterrey, Atizapán 52926, Mexico; 4Facultad de Estudios Superiores Zaragoza, Universidad Nacional Autónoma de México, Ciudad de México 09230, Mexico; 5Psychology Department, University of Los Andes, Bogotá 11605, Colombia; 6School of Engineering and Sciences, Tecnologico de Monterrey, Monterrey 64849, Mexico

**Keywords:** resilience to stress index, machine learning, physiological response, biofeedback, mindfulness, heart rate variability, academic stress management, Leaders of Tomorrow, higher education, educational innovation

## Abstract

In this work, we evaluate the effectiveness of a multicomponent program that includes psychoeducation in academic stress, mindfulness training, and biofeedback-assisted mindfulness, while enhancing the Resilience to Stress Index (RSI) of students through the control of autonomic recovery from psychological stress. Participants are university students enrolled in a program of excellence and are granted an academic scholarship. The dataset consists of an intentional sample of 38 undergraduate students with high academic performance, 71% (27) women, 29% (11) men, and 0% (0) non-binary, with an average age of 20 years. The group belongs to the “Leaders of Tomorrow” scholarship program from Tecnológico de Monterrey University, in Mexico. The program is structured in 16 individual sessions during an eight-week period, divided into three phases: pre-test evaluation, training program, and post-test evaluation. During the evaluation test, an assessment of the psychophysiological stress profile is performed while the participants undergo a stress test; it includes simultaneous recording of skin conductance, breathing rate, blood volume pulse, heart rate, and heart rate variability. Based on the pre-test and post-test psychophysiological variables, an RSI is computed under the assumption that changes in physiological signals due to stress can be compared against a calibration stage. The results show that approximately 66% of the participants improved their academic stress management after the multicomponent intervention program. A Welch’s *t*-test showed a difference in mean RSI scores (*t* = −2.30, *p* = 0.025) between the pre-test and post-test phases. Our findings show that the multicomponent program promoted positive changes in the RSI and in the management of the psychophysiological responses to academic stress.

## 1. Introduction

The concept of stress encompasses both psychological and physiological aspects, since it is integrated by the abstract perception of a demand from a real environment and the biological repercussions of subsequent adaptation [1,2]. Biologically, stress can be analyzed based on the fact that, from one individual to another, there will be variations in the physiological markers that are altered when experiencing a stressful situation. Among these markers, elements directly related to the activation of the Autonomic Nervous System, and more specifically its sympathetic component, stand out, such as the galvanic response of the skin, the blood volume pulse, and the frequency of breathing, among others [3]. For example, research conducted by Lizheng et al. [4], which aimed to understand which physiological markers can determine specific emotions, suggested that the identification of emotions is not obtained from a singular psychophysiological signal, but from the interaction of the observed behavior of several of them. Several studies have been conducted to observe the psychophysiological impact of anxiety or psychological stress in different scenarios. In these studies, stress has been induced using different psychological stressors either in the laboratory or in real scenarios, using machine learning models for the analysis of physiological responses such as peripheral temperature, electrodermal activity, and heart rate variability, among others. On the same line, Šalkevicius et al. in Ref. [5] recorded blood pressure, galvanic skin response, and distal peripheral temperature to determine their subjects’ anxiety levels during a training program for managing social anxiety disorder (fear of public speaking) using virtual reality applications.

Due to the psychophysiological characteristic of stress, the ability to recover from alterations generated by the adaptation to perceived demands has both mental and biological implications [6]. In this study, the definition of resilience to stress is considered as the “properties contributing to the speed and amount of possible recovery of physiological variables after exposure to a stressful event” [7].

In this work, we evaluate the effectiveness of a multicomponent program in university students with academic excellence, aiming at improving academic stress management. The multicomponent program includes psychoeducation in academic stress, mindfulness training, and biofeedback-assisted mindfulness; the invited students participate in the “Leaders of Tomorrow” scholarship program of Tecnológico de Monterrey University.

To evaluate the effectiveness of the program, we used a recently proposed index to measure the resilience to stress of an individual, based on his/her physiological response to stressful situations. The index was proposed by Díaz et al. [7] and is referred to as Resilience to Stress Index (RSI). Our results indicate that the program enhanced the RSI of participants through the control of autonomic recovery from psychological stress.

### 1.1. Elements of the Multicomponent Program Proposed in this Work

Next, we briefly describe the intervention program techniques applied in this research.

#### 1.1.1. Mindfulness

There is a diverse range of techniques and interventions that have been used for the development of coping strategies for stress. Psychoeducation about stress, understood as a therapeutic approach that enhances therapeutic compliance through informed decision-making about stress management, such as its symptoms or risk and protective factors, is a valuable component in programs targeting stress, and it is recommended as part of multimodal interventions [8,9]. Physical exercise, relaxation techniques, biofeedback, and mindfulness training, are other approaches that have proven efficacy in the management of stress [8,10]. Mindfulness is to intentionally pay attention or become aware of what is happening in the present moment, with acceptance and in a non-judgmental way (non-judgmental awareness) [11]. Mindfulness-based programs train awareness through formal meditation practices that include different types of meditation, such as breathing meditation and body scan meditation, among others, as well as informal meditation practices that are done during activities of everyday life [12].

#### 1.1.2. Biofeedback

Biofeedback is a technique for developing self-regulation strategies that increase the voluntary control of physiological (related to the autonomic nervous system) and cognitive processes. Through this technique, the interaction between the sympathetic and parasympathetic systems, as well as the autonomic reactivity and recovery from stress are recorded. For example, maintaining controlled diaphragmatic breathing with biofeedback results in an increased heart rate variability (HRV), decreased blood glucose, and baroreflex activity with sympathetic-vagal balance, respiratory sinus arrhythmia, promoted resonance frequency, and improved cardiac autonomic function [13,14,15].

#### 1.1.3. Biofeedback-Assisted Mindfulness

Combining mindfulness and biofeedback in the same intervention enhances the ability to pay attention to the present moment and physical sensations, which is achieved typically with mindfulness training. Mindfulness strengthens the skill of non-judgmental awareness that favors biofeedback training by avoiding thoughts of rejection or control that activate the sympathetic system [16]. On the other hand, biofeedback makes it possible for the subject to know their physiology through instrumental measurements that provide information during skill training for subjects to observe their “inner world”, the physiological changes that happen and how their thoughts, movements, emotions, and actions affect it, making it possible to self-regulate the arousal of the autonomic nervous system during non-judgment skill training [16,17,18,19,20,21,22].

The primary physiological response used in biofeedback as an addition to mindfulness is HRV because of its effectiveness in self-regulation training [23]. With HRV biofeedback, it is possible to learn self-regulation skills that mindfulness training alone does not offer. For example, the benefit of mindfulness breathing with biofeedback is enhanced by accompanying it with a balance in autonomic nervous system functioning. In short, mindfulness enhances the ability to be aware of physical sensations, while biofeedback helps regulate autonomic nervous system arousal [20].

Some of the benefits of the combination of mindfulness and biofeedback are the improvement of emotional self-regulation, empathy, compassion, a decrease in distress and anxiety, as well as positive changes in physiological indicators (increased HRV, changes in blood pressure and respiratory rate) and biochemical markers (HbA1c, cortisol, and triglycerides) [24]. Furthermore, biofeedback training and practice allow participants to have control over their autonomic responses [20,25].

Despite the existing evidence about the effectiveness of both techniques for stress management, to the best of our knowledge, in Mexico there are no protocols that apply mindfulness and/or biofeedback-assisted mindfulness techniques to enhance academic stress management. Hence, it is essential to design and implement intervention strategies that can prevent and address the psychophysiological outcomes of academic stress by teaching the students about stress, its symptoms, and the psychophysiological implications that could eventually trigger a physical disease (e.g., hypertension) or a psychological disorder (e.g., generalized anxiety).

As we will see later in the related work section, there is a lack of a standard way to evaluate the success of a mindfulness or equivalent program. In our study, we propose the RSI as a metric for fair comparison purposes. Therefore, the contributions of this work are the following:1.A novel multicomponent program specially designed to improve stress management;2.The use of a new index, the RSI, to measure the effectiveness of such a multimodal program.

This paper is organized as follows: In Section 1, we presented our introduction, including the main techniques employed during the study. In Section 2, we introduce related work relevant to ours. In Section 3, our methodology is explained in detail along with Section 4, where we describe our procedure. In Section 5 we present our final results and in Section 6 we give a discussion, limitations of the study, and future lines of research.

## 2. Related Work

### 2.1. Regarding Known Physiological Markers Closely Related to Stress

Concerning the galvanic response of the skin, several studies have found that there is an increase in sweating when experiencing a stressful situation [26]. The greater the sweating, the higher the skin conductance (SC), so there is a positive correlation between SC and the stress experienced by a person, and this happens especially in the limbs, armpits, and face [27]. Generally, sweating measurements are performed based on skin conductance using micro Siemens units.

One of the most evident bodily effects of stress is the increase in heart rate. Generally, heart activity is deeply involved with the autonomic nervous system, as well as regulatory vagal and circadian rhythm modulation. The sympathetic nervous system makes it easier for the heart to beat with greater force and speed, with the main function of providing the body with oxygenated blood to meet the perceived needs of the environment [28]. Consequently, the activity of the heart related to stress is deeply involved in the breathing process. A measurement of Heart Rate Variability (HRV), found through the analysis of the variability of time intervals between heartbeats, can be used to determine the effects of stress on the body. Several studies have found that stressful experiences lead to a lower HRV [29,30]. Likewise, blood volume pulse (BVP), despite not having a standard unit, can be used to calculate the HRV, since there is a measurable signal change when sympathetic arousal is detected, shown in the variability of intensity between signal peaks.

Another reaction of our body strongly related to stress is the frequency of breathing, considering that the activation of the cardiovascular system fosters an increase in the frequency of breathing to function effectively through the pumping of oxygenated blood [31]. Thus, stress-derived body activation implies a higher breathing frequency in contrast to a relaxed body state [32].

### 2.2. Regarding Stress and Resilience to Stress

Several authors agree that the main source of stress in adolescents and young adults is academic performance [33], even reporting a prevalence of stress in 55% to 87.9% of students [34]. The repercussions of stress in this age group are observed in university students who frequently encounter academic challenges and responsibilities that, when not adequately addressed, may result in emotional and behavioral changes, health disorders, and school difficulties [35]. Studies in the school environment have suggested that academic stress occurs at all school levels, but that upon reaching university, it is at its highest peak due to both heavy workloads and their complexity, as well as the fact that they coincide with a stage in life where the student faces decisions that will define the following years of his or her life [36,37]. Moreover, entering and staying in college potentially coincides with the process of separation from the family into independence, incorporation into the labor market, romantic relationships, and other highly stressful decisions [38].

In recent years, several studies have focused on the feasibility of enhancing stress resilience in different contexts. In 2019, in a study by Kloudova et al. [39], aviation pilots underwent mental training biofeedback therapy to reduce physiological stress symptoms, which are indicators of anxiety that could impede flying performance. The measurements included physiological stress-related variables such as BVP, heart rate, HRV, SC, temperature, and breathing rate. After six biofeedback sessions, a *t*-test evidenced an important contrast of *p* < 0.01 between pre-test and post-test measurements. Vitasari et al. in 2011 worked with young adult university students with anxiety-related symptoms, which are importantly similar to stress response due to the activation of the sympathetic system. They conducted a biofeedback training program that used heartbeats per minute, as well as breaths per minute as measurements [40]. They concluded that, after 10 training sessions, psychophysiological management of bodily responses was successful in gaining control of symptoms of anxiety.

### 2.3. Regarding the Multicomponent Program Elements

Mindfulness training has been successfully implemented in a variety of populations and for diverse objectives. A recent systematic review of 44 meta-analyses evaluated mindfulness-based interventions (MBIs) in populations that included adults, students, medical professionals, and children for problems such as anxiety, eating disorders, depression, pain, stress, and different physical and health issues. They concluded that MBIs have significant effects with transdiagnostic relevance except for substance use and sleep disorders [41]. Many examples of successful implementation of mindfulness training can be found in scientific literature. In diabetes patients, for example, results showed better emotional regulation, acceptance and reinterpretation of thoughts, more adaptive behaviors, reduction of distress at the hypothalamic pituitary adrenal axis level, and improvement of diabetes self-efficacy, and metabolic control [42,43,44]. Relevant to stress coping, the mechanisms of mindfulness include the ability to direct mental resources to the development of behaviors that are under the control of the subject and to discern between alternative responses instead of focusing strictly on the stressful event [21].

In education, the implementation of mindfulness in college students resulted in the improvement of working memory, attention, academic performance, social skills, emotional regulation, self-esteem, mood, and the reduction of anxiety, stress, and fatigue. Positive effects have also been reported in the areas of critical thinking, focus, test anxiety, test scores, study habits, organizational skills, self-control, and attention deficit hyperactivity disorder [45]. Additionally, the combination of mindfulness and biofeedback has been successful in reducing stress, improving academic performance, and training self-regulation skills in both students and teachers [17,18,19,23]. Rush et al. (2017) applied mindfulness and biofeedback to 14 students diagnosed as emotionally disturbed, and in comparison to a control group, achieved significant changes (t (29) = 2.730, *p* = 0.011) with a large effect size (d = 0.985; 95 CI = 0.236, 1.73) specifically in reducing disruptive behaviors unrelated to academic tasks, which would be expected to have a positive impact on academic performance [19]. Biofeedback was provided through an electronic game to teach and guide proper breathing rate to improve HRV.

## 3. Methodology

We employed a quasi-experimental design pre-test and post-test by following the steps below.

1.We measured autonomic recovery from psychological stress through a stress profile that includes the simultaneous recording of four physiological responses (heart rate variability, blood volume pulse, breathing rate, and galvanic skin response), using the Procomp Infiniti biofeedback equipment;2.We applied the multicomponent program integrated by psychoeducation of academic stress, training in mindfulness, and biofeedback-assisted mindfulness;3.Finally, we compared the RSI of participants obtained during the pre-test against the one attained during the post-test.

### 3.1. Participants

Participants were an intentional and voluntary sample of 38 undergraduate students with high academic performance, 71% (27) women, 29% (11) men, and 0% (0) non-binary, with an average age of 20 years (18 to 25), belonging to the “Leaders of Tomorrow” scholarship program from Tecnológico de Monterrey University in Mexico. The inclusion criteria were that participants did not have any heart disease or were on anxiolytics or antidepressant medication. When any of these conditions occurred, they had to report it during the initial interview; as a result, they were removed from the study even though they could continue participating if they wished. However, their results were not taken into account for the final report.

### 3.2. Equipment and Data Acquisition

The psychophysiological measurements during the stress profile and the biofeedback-assisted mindfulness sessions were recorded using a laptop and a ProComp5 Infiniti equipment model T7525 (https://thoughttechnology.com/procomp5-infiniti-system-w-biograph-infiniti-software-t7525/, accessed on 25 January 2022), which is integrated by a 4-channel decoder that measures heart rate variability, blood volume pulse, breathing rate, and galvanic skin response. Non-invasive surface electrodes are used to record these responses. The ProComp5 device has an ADC (Analog to Digital Converter) resolution of 14 bits, and we measured using 256 samples per second (256 Hertz sample frequency). Table 1 shows the description of variables measured with the biofeedback device. It is worth noting that HRV is obtained through BVP recording, so it is measured through the same sensor.

Figure 1 illustrates the raw data of a single subject during the ProComp5 measurements performed at different phases of the study. The three variables are represented in each slot through time (256 samples per second). Table 2 presents the descriptive statistics of such data.

### 3.3. Signal Preprocessing

The next two sections describe the data preprocessing performed on the subject’s data obtained using the biofeedback device.

#### 3.3.1. Median Filter

To avoid pseudo detection of peaks or noise, the acquired signals must be preprocessed before using the data. Thus, we applied a median filtering technique to remove the peaks and noise from the signal. The kernel size *w* of the filter was selected according to Equation (Equation 1) [46].
(1)$$w=\frac{1}{4f_{s}}*length\left(n\right)$$
where $f_{s}$ is the sampling frequency and length(n) is the total number of instances. Moreover, the raw signals from the biofeedback device have an offset at the beginning of the sensor measurements; therefore, for our analysis, we removed the first 0.5 s of data.

#### 3.3.2. Standard Scaler

The unit of measure of each sensor is on a different scale; hence, a critical step is standardization. For this purpose, we applied a standard scaler after the median filter. We used Equation (Equation 2) to standardize each point of all features, where *a* is the mean of the feature, and *s* is the standard deviation of the variable. Standard scaling is a way of normalizing features by deleting their mean and scaling their variance to one. Since the normalized value is determined uniquely by the mean and variance, it has some advantages, including being linear, reversible, rapid, and highly scalable [47].
(2)$$z=\frac{x-a}{s}$$

Figure 2 illustrates the laboratory setup. (A) Psychophysiological stress profile measurement during the pre-test and post-test evaluation sessions (see Section 4.1 and Section 4.3). (B) Biofeedback-assisted mindfulness training session (see Section 4.2). (C) Sensor placements as recommended by the manufacturer.

## 4. Procedure

The program consisted of 16 individual sessions during 8 weeks, divided into 3 phases: pre-test evaluation, multicomponent intervention program, and post-test evaluation. Figure 3 shows a diagram that represents the general procedure of data collection.

### 4.1. Phase 1: Pre-Test Evaluation

This phase was carried out in the first week in a 30-min in-person session. During this phase, the psychophysiological stress profile of the participant is built. At the beginning of the session, non-invasive surface electrodes are placed on the participant to measure four physiological responses in real-time: heart rate variability, blood volume pulse, breathing rate, and galvanic skin response. After the lab setup, the psychophysiological stress profile is recorded, with a duration of 18 min, divided into 9 stages of 2 min each. The first stage consisted of a baseline measurement, in which the participant is instructed to remain silent and with her/his eyes closed. Then, the subsequent stages alternated between stressful tasks, where the participant is presented with different psychological stressors, and recovery periods. The stressful tasks are: a Stroop test, an arithmetic test (such as subtracting 7 by 7 from a random four-digit number), an auditory stressor, and an emotional stressor in which the participant evokes a stressful academic moment. Figure 4 shows the activities of the psychophysiological stress profile creation.

### 4.2. Phase 2: Multicomponent Intervention Program

The following subsections describe the elements of the multicomponent intervention program: psychoeducation on academic stress, mindfulness training, and biofeedback-assisted mindfulness.

#### 4.2.1. Psychoeducation on Academic Stress

This is an online task carried out asynchronously in one session (session 2) during week two. The session included pre-recorded videos that addressed various topics, such as the conceptualization of academic stress, sources of stress, symptomatology, risk factors, protective factors, and coping techniques.

#### 4.2.2. Mindfulness Training

This training spans 2 weeks (weeks 3 and 4), including 10 online sessions (sessions 3 to 12), with 1 per day. Each session included a video explaining the topic and an audio guide detailing a body scan and breathing awareness meditation.

The training begins with basic concepts and scientific foundations of mindfulness; it also explains the conceptualization and scientific grounds of mindfulness training. Likewise, it includes information and practices on awareness of language, emotions, thoughts, and physical sensations, as well as acceptance, non-judgmental awareness, and the application of mindfulness to stress management.

#### 4.2.3. Biofeedback-Assisted Mindfulness

This task consists of three in-person sessions (sessions 13–15) taking place in a 2-week period, considering at least 1 session per week. The sessions begin by placing surface sensors on the participant, to record in real-time HRV from blood volume pulse and diaphragmatic breathing. The training session lasts 24 min, divided into 6 stages. The first stage lasts for two minutes, during which physiological measurements are recorded, which serve as a baseline. In stages two and four, the participant listens to audio guides of mindfulness meditations: breathing awareness (Audio 1) and body scan (Audio 2), each lasting five minutes. In stages three and five, also five minutes each, participants continue the meditation with their eyes open, and are asked to observe a computer screen that provides biofeedback of their HRV; the screen plays a video that only moves forward when the participant manages to increase their HRV. Lastly, during stage six, for two minutes, physiological measurements are recorded at rest. It is expected that participants develop control of their physiological responses as an outcome of practicing mindfulness meditation. Figure 5 shows the structure of the biofeedback-assisted mindfulness sessions.

### 4.3. Phase 3: Post-Test Evaluation

During this phase, the post-test psychophysiological stress profile of the participant is built. It takes place during 8 eight, session 16, to close the intervention program. The procedure is carried out in the same way as in Phase 1, described in Section 4.1.

It is important to mention that the protocol, considering mainly the psychophysiological stress profile performed in phase 1, and the biofeedback component of the biofeedback-assisted mindfulness sessions, performed in phase 2, was based on the Mindfulness Suite by the Biofeedback Federation of Europe (https://www.bfe.org/buy/mindfulness-suite-p-550.html, accessed on 25 January 2022), translated into Spanish and adapted for the target population.

## 5. Experimental Results

After a quick introduction to Principal Component Analysis, we briefly introduce the resilience to the stress index used in this work, and then describe how we employed the index to measure the effectiveness of the multimodal intervention program.

### 5.1. Principal Component Analysis

The Principal Component Analysis (PCA) is a well-known unsupervised learning technique for reducing the dimensionality of data. Moreover, it increases interpretability and, at the same time, minimizes information loss. Additionally, it transforms the data, making it easier for 2D and 3D visualization [48].

We present Figure 6 only for visualization purposes. The figure shows the distribution of clusters using two principal components of the PCA of a single subject. As it can be noticed, clusters are well-defined at each stage. It is worth mentioning that for the rest of the experiments, we use all principal components.

### 5.2. Resilience to Stress Index (RSI)

The Resilience to Stress Index (RSI) is a new indicator proposed by Díaz et al. [7] to measure a subject’s ability to recover from stress. The index is calculated starting from a “baseline” or calibration stage, obtained by measuring the vital signs of each person before starting with the application of emotional stressors. During the following stages, the evaluator intervenes with stressors and successively provides a quiet space. In this way, changes in the users’ vital signs, as well as their attempts to return to their baseline, are recorded. It is important to note that the baseline, as well as the other stages, are unique to each participant.

With the Principal Component Analysis technique, each stage is distinguished by having its own cluster corresponding to the psychophysiological measurements; in other words, it is possible to obtain the distances between clusters. PCA obtains a covariance matrix that eliminates the multicollinearity between its attributes [49], making it possible to calculate the Euclidean distance and use it as a valid metric to represent these distances. Equation (Equation 3) is used to compute the RSI for each subject.
(3)$$\mathrm{RSI}=\frac{\Delta R}{\Delta S}$$

For this equation, ΔR represents the distance between the centroids of the last two stages, which in this case, are eight and nine. ΔS represents the largest distance found between the centroid of the baseline and the other phases given for the sample.

As a result, the RSI is able to capture the physiological responses of each individual. An RSI close to one indicates that the individual has a high resilience to stress by presenting the ability to recover from stressors and approaching his/her initial baseline. Small values, on the other hand, indicate that the individual failed to return to the baseline and showed poor resilience.

### 5.3. Use of the Resilience to Stress Index (RSI) in Our Study

To compute the RSI, we followed the methodology presented in Ref. [7]. For each subject in the dataset, based on the PCA components, we used the Euclidean Distance to calculate inter-stage distances, and their RSI using Equation (Equation 3).

As mentioned previously, we used the Procomp5 biofeedback device to obtain the physiological variables during the psychophysiological pre-test, and post-test sessions. For each subject, we calculated the RSI at each session. In Table 3, we present our results obtained during the pre-test and post-test sessions.

Figure 7 shows the box and whisker diagram of the data with the quartile values shown in Table 4. Furthermore, we point out a couple of interesting things, as all points in Q4 of the post-test are higher than the upper whisker of the pre-test, except for the outliers, and the median in the post-test (0.28) is higher than the median in the pre-test (0.22).

The results show that approximately 66% of the participants (80% women and 20% men) improved their stress management after applying the multicomponent intervention program, which we consider encouraging, due to the short intervention time and the psychological complexity of stress management in our current time and society. The means and standard deviations of the RSI values for the pre-test and post-test are ${\bar{X}}_{1}=0.247,s_{1}=0.096$ and ${\bar{X}}_{2}=0.321,s_{2}=0.174$, respectively, showing an improvement in the overall performance of participants.

A Welch’s *t*-test was performed to determine if there was a statistically significant difference in RSI scores obtained during the pre-test and post-test. Between tests, participants were exposed to an intervention program (psychoeducation in academic stress, mindfulness training, and biofeedback-assisted mindfulness). The sample size for both groups was 38 students. The Welch’s *t*-test rejected the null hypothesis and revealed a difference in mean RSI scores (*t* = −2.30, p=0.025) between the two groups. Table 5 summarizes the metrics gathered to compute the statistical test assuming unequal variances.

To reinforce this conclusion, we decided to run a non-parametric hypothesis test, such as Wilcoxon signed-rank test on the data. The results confirmed the statistical significance, as shown in Table 6, where the null hypothesis, i.e., the two samples are equally distributed, is rejected.

## 6. Discussion

In addition to learning and academic challenges, the Leaders of Tomorrow academic excellence students experience stressors at school that arise from many sources, including possible disruptions to the family system, peer conflicts, maintaining a scholarship, socio-cultural components, and vulnerability to physical and mental health risk factors. Depending on individual characteristics, these stressors may represent a challenge, a learning opportunity, or negatively affect academic and personal life.

The results of this research showed a favorable RSI in students who participated in the multicomponent biofeedback-assisted mindfulness program, which meant adequate coping and resilience in the face of academic stress. The RSI is derived by comparing changes in physiological signals due to stress against a calibration stage, and was first introduced by Díaz et al. in Ref. [7]. The current research is innovative in Mexico since, to the best of our knowledge, there is no precedent of a multicomponent program that successfully integrates psychoeducational elements, mindfulness, and biofeedback focused on self-management of academic stress and its autonomic modulation in university students of academic excellence. The RSI obtained from an objective and precise protocol to measure physiological indicators during psychophysiological stress is an accurate measure to determine the changes in the autonomic nervous system in the face of sympathetic activity produced by cognitive or perceptual tasks compared with periods of recovery or production of parasympathetic activity in university students. Depending on the result obtained from each participant, the RSI can be considered as a protective (increased HRV) or health risk (decreased HRV) differentiating factor that can impact the student’s health-disease process [6]. The results of this study show that 66% of students significantly improved their stress resilience.

Looking at possible reasons for this finding from the psychological mechanism perspective, there are some meaningful connections to make between this finding and what has been found in the literature. Chin and colleagues published in 2019 a study in which they investigated the role of acceptance training inside the Mindfulness Based Stress Reduction (MBSR) program as a mechanism of action for stress resilience [50]. Their findings suggest that the training in acceptance that takes place in MBSR through the instruction of meeting what is experienced during meditation with an attitude of acceptance, is a necessary component that leads to stress resilience, resulting in training of acceptance as a skill. The multicomponent program used in our research included acceptance training in two ways: first, by including acceptance as a topic in the mindfulness training component and second, by repeating the phrase “...with acceptance and non-judging attitude...” during the meditation guides, including the guides used for the biofeedback-assisted mindfulness sessions. Acceptance training through mindfulness has also been mentioned as an essential component of biofeedback training in stress reduction [16], as discussed earlier in the introduction of this work.

Along with mechanisms of action, Steffen and Bartlett gathered information from empirically supported interventions addressing stress resilience in both biofeedback and psychotherapy and so, pointing to three specific mechanisms or, what they call practices, to build stress resilience as being: balancing of life demands with equanimity, awareness leading to the reduction of worry, and engaging in flexible coping skills [51]. In connection with acceptance training, they include acceptance as an important component for awareness to reduce worry [16,50]. It is clear that the combination of biofeedback and mindfulness in our proposed multicomponent program contributed to the training of awareness of stressful tasks and of the physiological manifestations of stress along with acceptance training, and at the same time, provided the opportunity of experiencing the use of mindfulness meditation as a coping skill to use for stress reduction.

It has been stated in biofeedback literature that the use of HRV biofeedback for stress reduction is based on the fact that higher HRV values are associated with individuals being more physically and emotionally resilient [52], resulting in an increased ability to respond more skillfully during stressful tasks. During training sessions in phase two of this study, most of the students were able to play the video, meaning the participants raised their HRV level during the meditation practice; additionally, they were simultaneously informed about the effect of meditation on their physiology and training emotional resilience. This is different from what is usually done in mindfulness training for stress reduction, in which participants do not know about the impact of meditation at the moment, but further when they start to notice changes in their stress responses in everyday experiences [53].

Whether the addition of biofeedback can accelerate the training on stress resilience due to objectively learning about changes in stress physiology and achieving higher HRV values, is a very interesting topic that entails future research.

The results of this research allow us to observe how with sustained mindfulness practice, college students improved attentional and emotional self-regulation and its effect on mindfulness psychophysiology to manage academic stress as well as to gain potential health benefits (e.g., immune system function, cardiovascular, neuroendocrine, improved health behaviors including eating, sleeping, etc., more positive moods, improved quality of life, and increased gray matter density in the hippocampus) [20,54].

Our results contribute to knowledge and evidence-based practice that seeks to integrate clinical psychological research preventing health problems from a psychosocial perspective in academically outstanding university students [40]. Psychophysiological assessment procedures and the multicomponent program for self-management of academic stress offer a viable option to comprehensively address (psychological and psychophysiological) the eventual clinical symptomatology reported by these students [45]. A detail worth noting is that the current research was carried out throughout a regular academic semester. The timing of the post-test was effectively close to the final academic evaluations of the participants. Usually, the evaluations at the end of the school term tend to be one of the most stressful moments for a student, due to factors such as the accumulated psychological tension from the progression of the term or the perceived impact of the evaluations on the final grade [55]. Although this context may have altered the results obtained, it is suggested that, despite the fact that the students were in a naturally more stressful environment, a significant decrease in stress was achieved. Conducting this research during a regular academic period allowed a natural evolution of stress to occur in these students. However, one of the limitations of this research is the absence of a control group that reveals the differences in autonomic activity and psychological stress between students of the Leaders of Tomorrow scholarship that took part in the multicomponent program and those who did not participate. Likewise, since the participants are part of a high-performance scholarship program, the sample may not be representative of all university students. This research was designed and conducted during the COVID pandemic, which limited the inclusion of a control group or more university participants to strengthen the internal validity of the results, especially considering a more balanced gender distribution. Despite said limitations, the results reported allowed for the observation of favorable changes in the RSI of participants in the program.

Certain institutions provide technological devices for both academic activities and personal use [56], and the inclusion of biometric devices that encourage practices such as HRV enhancement training through biofeedback could have a positive impact on the development of resilience to stress in an academic context. However, based on the results shown in this research, it is recommended that schools incorporate in their educational policies the practice of mindfulness assisted with biofeedback, targeting teachers, students, administrative, and managerial staff. The goal would be to improve their capacity to face their psychosocial and academic challenges, according to the scientific evidence based on the effectiveness of this type of program [18,19,23]. In the case of students, specific interventions can be conducted for groups vulnerable to anxiety, attention deficit disorder, academic lag, among other problems.

It is also recommended to continue with the psychophysiological assessment protocol used in this research as it will allow monitoring, throughout their academic life at the university, the psychological adjustment of students of academic excellence by detecting and addressing moments vulnerable to academic stress and thus improving their psychological well-being. The current research focused on high-achieving students who must maintain high performance in order to preserve their scholarship. Likewise, the incentive received by the participants was social service hours, which are also part of their scholarship requirements. Future work regarding this line of research includes but is not limited to: (a) adding a control group to assess the effectiveness of the program within the same population, which would strengthen the internal validity of this research; (b) replicating the experiment in a representative sample of university students, considering a distribution that takes into account the variability of bodily responses to stress in regard to gender, age, and other demographic variables; (c) reducing the number of sessions in the program to make it more accessible to the academic activities of the participants, which would increase student interest in participating in the program without being distracted from their own academic or personal activities; (d) including more face-to-face activities that allow timely feedback on the skills acquired through the program as well as an immediate correction of meditative practices; (e) carrying out follow-up sessions to measure the permanence of the favorable results obtained in the post-test, strengthening the evidence on the acquisition and development of a resilience to stress skill; (f) broadening the profile of participants to consider other characteristics, such as students with low or mixed academic performance, those that are part of a representative cultural or sports team, students that also have jobs, or those that have a diagnosed mental disorder, especially one related to poor stress management; (g) considering a statistically relevant and balanced sample to address gender differences in stress resilience, which could develop into a research path that puts emphasis on the physiological stress response based on gender. 

## Figures and Tables

**Figure 1 sensors-23-02650-f001:**
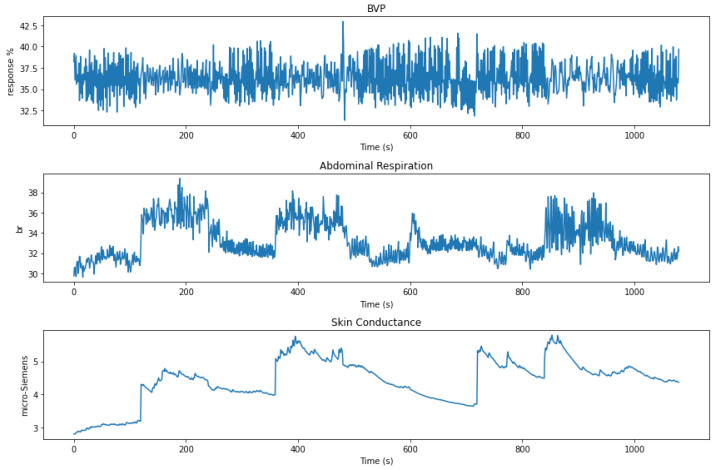
Raw signal of the three physiological variables of subject 14 (woman, age 21), during the ProComp5 measurements.

**Figure 2 sensors-23-02650-f002:**
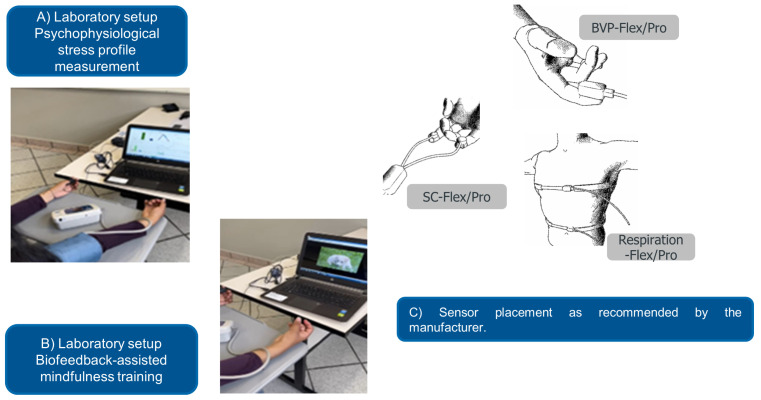
Laboratory setup. (**A**) Psychophysiological stress profile measurement. (**B**) Biofeedback-assisted mindfulness training session. (**C**) Sensor arrangement as suggested by the manufacturer. Images taken from the ProComp5 InfinitiTM Hardware Manual with permission of Thought Technology, Montreal, QC, Canada. www.thoughttechnology.com (accessed on 21 December 2022).

**Figure 3 sensors-23-02650-f003:**
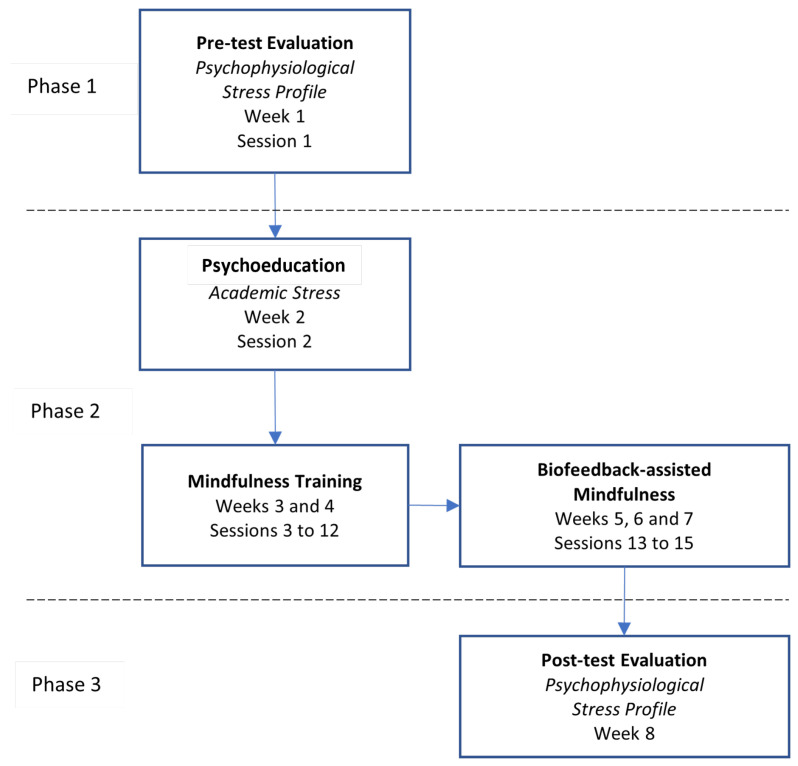
Diagram showcasing the multicomponent program and data collection procedure.

**Figure 4 sensors-23-02650-f004:**
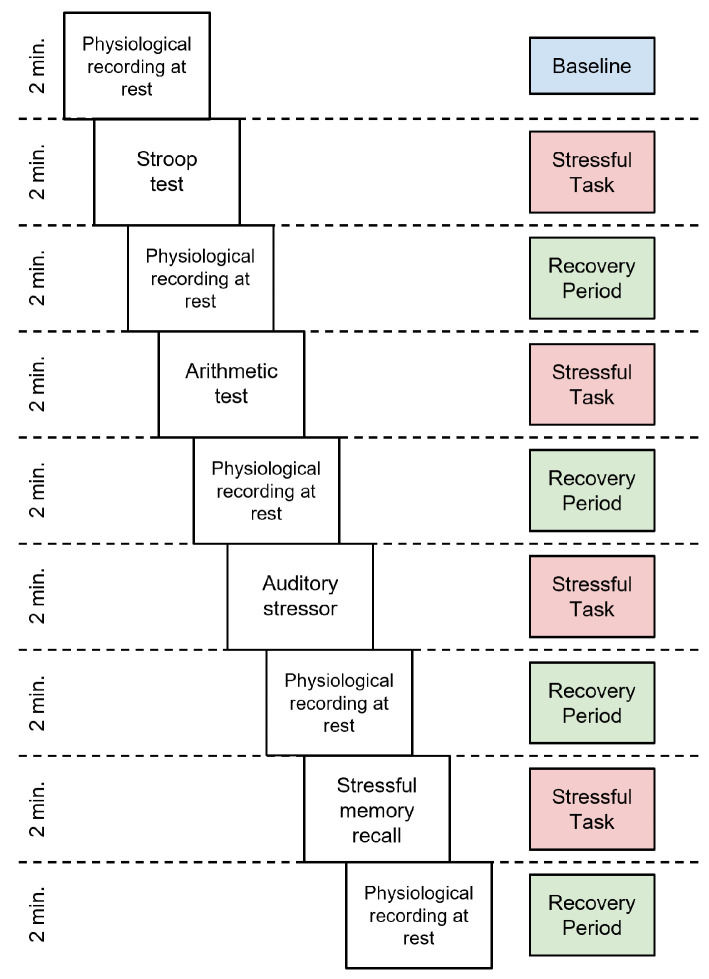
Chronological representation of the psychophysiological stress profile.

**Figure 5 sensors-23-02650-f005:**
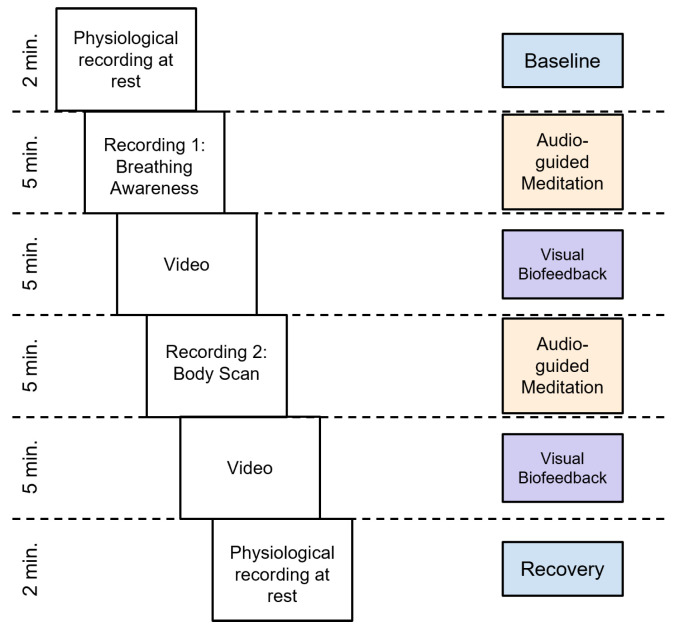
Chronological representation of the biofeedback-assisted mindfulness training session.

**Figure 6 sensors-23-02650-f006:**
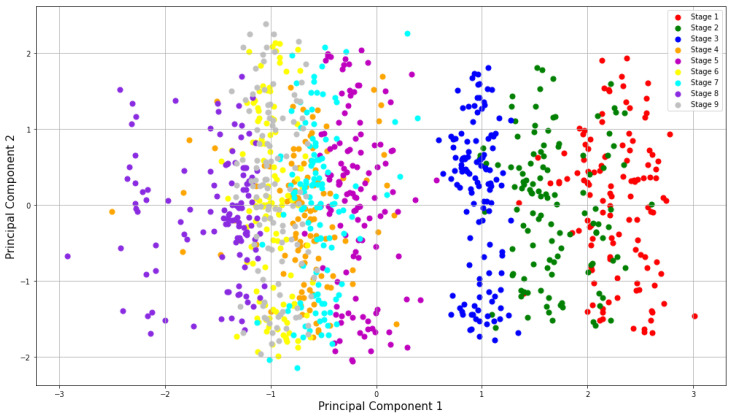
2D visualization using the 2 principal components of subject 23 during the pre-test session.

**Figure 7 sensors-23-02650-f007:**
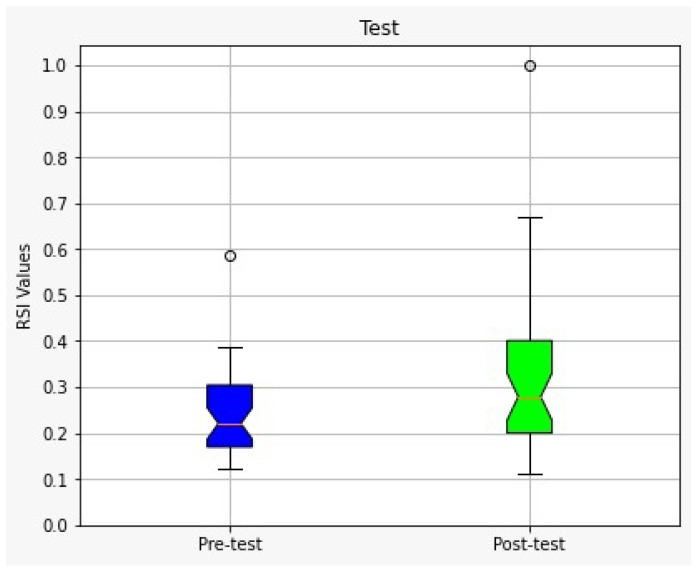
Boxplot diagram corresponding to the pre-test and post-test performance of all subjects.

**Table 1 sensors-23-02650-t001:** Description of variables measured with the biofeedback device.

	Skin Conductance (SC)	Blood Volume Pulse (BVP)	Heart Rate Variability (HRV)	Breathing Rate (BR)
Description	Measures of the peripheral skin conductance	Measures of changes in the arterial translucency	Temporal variation in heart rate or inter-beat interval	Measures of the expansion and contraction of the rib cage
Sensor	SC-Flex/Pro	BVP-Flex/Pro	BVP-Flex/Pro	Respiration-Flex/Pro
Unit of Measure	micro-Siemens	Unit less quantity displayed as percentage	Variation in heartbeats measured in milliseconds	Unit less quantity displayed as percentage
Range of values	0 μS–30 μS	0–00%	Can oscillate between 20 and over 200 milliseconds	0–100%
Typical Relaxed Measures	2 μS	30–60%	No typical measure	No typical measure

**Table 2 sensors-23-02650-t002:** Descriptive statistics of the signals from three physiological variables obtained during the ProComp5 measurements. They describe the signals shown in Figure 1.

	Blood Volume Pulse	Abdominal Respiration	Skin Conductance
Mean	36.329	33.227	4.420
Standard Error	0.004	0.003	0.001
Median	36.162	32.678	4.521
Mode	35.813	32.263	4.090
Standard Deviation	1.969	1.836	0.671
Sample Variance	3.876	3.370	0.451
Kurtosis	25.239	−0.429	−0.166
Skewness	−1.210	0.693	−0.522
Range	50.333	10.604	3.012
Minimum	0.004	29.509	2.797
Maximum	50.337	40.113	5.809
Sum	10,044,141.52	9,186,504.78	1,222,087.534
Count	276,480	276,480	276,480

**Table 3 sensors-23-02650-t003:** Pre-test and post-test RSI for each subject. (Difference = RSI (post-test) − RSI (pre-test)).

Subject	RSI	RSI	Difference	Subject	RSI	RSI	Difference
	(Pre-Test)	(Post-Test)		(Continue)	(Pre-Test)	(Post-Test)	
S1	0.124	0.326	+0.202	S20	0.169	0.492	+0.323
S2	0.180	0.148	−0.032	S21	0.255	0.297	+0.041
S3	0.217	0.369	+0.152	S22	0.367	0.164	−0.202
S4	0.136	0.112	−0.024	S23	0.388	0.327	−0.061
S5	0.381	0.407	+0.025	S24	0.336	0.169	−0.167
S6	0.305	0.273	−0.032	S25	0.165	0.410	+0.246
S7	0.267	1.000	+0.733	S26	0.585	0.324	−0.261
S8	0.185	0.134	−0.051	S27	0.301	0.273	−0.028
S9	0.221	0.274	+0.053	S28	0.259	0.343	+0.083
S10	0.183	0.201	+0.018	S29	0.335	0.384	+0.049
S11	0.310	0.223	-0.087	S30	0.200	0.526	+0.326
S12	0.148	0.161	+0.014	S31	0.168	0.456	+0.288
S13	0.349	0.120	−0.229	S32	0.221	0.274	+0.053
S14	0.133	0.463	+0.330	S33	0.210	0.481	+0.271
S15	0.161	0.172	+0.010	S34	0.299	0.220	−0.079
S16	0.199	0.359	+0.160	S35	0.161	0.254	+0.093
S17	0.271	0.200	−0.071	S36	0.269	0.284	+0.015
S18	0.241	0.671	+0.430	S37	0.145	0.169	+0.024
S19	0.368	0.515	+0.147	S38	0.173	0.231	+0.058
				Mean	0.247	0.321	—
				SD	0.096	0.174	—

**Table 4 sensors-23-02650-t004:** Boxplot diagram values.

	Lower Whisker	Lower Quartile	Median	Upper Quartile	Upper Whisker
Pre-test	0.12	0.17	0.22	0.30	0.39
Post-test	0.11	0.20	0.28	0.41	0.67

**Table 5 sensors-23-02650-t005:** Welch’s *t*-test: Two-sample assuming unequal variances.

	RSI	RSI
	Pre-Test	Post-Test
Mean	0.24693707	0.32112398
Variance	0.0092543	0.03026685
Observations	38	38
Hypothesized Mean Difference	0	
df	58	
t Stat	−2.3004049	
P(T ≤ t) one-tail	0.0125205	
t Critical one-tail	1.67155276	
P(T ≤ t) two-tail	0.025041	
t Critical two-tail	2.00171748	

**Table 6 sensors-23-02650-t006:** Wilcoxon signed-rank test of pre-test against post-test.

Comparison	$R^{+}$	$R^{-}$	Hypothesis (α=0.05)	*p*-Value
Pre-test vs. Post-test	234	507	Rejected	0.04776

## Data Availability

The data presented in this study are openly available in the following GitHub repository https://github.com/isabellanoriegaq/Resilience-to-Stress-Index.

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
