# Peer review of "Measuring the Effectiveness of a Multicomponent Program to Manage Academic Stress through a Resilience to Stress Index"

_sensors, 2023, doi:10.3390/s23052650_

Round 1
Reviewer 1 Report
The research article presents a study evaluating the effectiveness of a multicomponent program aimed at enhancing the Resilience to Stress Index (RSI) of university students through psychoeducation, mindfulness training, and biofeedback-assisted mindfulness. The study uses an intentional sample of 38 undergraduate students with high academic performance, and the program is structured over an eight-week period. The results show that approximately 66% of the participants improved their academic stress management after the multicomponent intervention program, with a statistically significant difference in mean RSI scores between pre-test and post-test phases.
Overall, the study is well-conducted, with a clear research question and a well-defined methodology. The use of a psychophysiological stress profile to measure changes in RSI is appropriate, and the results are promising.
However, the presented research must be contextualized better by discussing a related research on emotion recognition (doi:10.5755/j01.itc.49.3.23948, doi:10.5755/j01.itc.51.1.29430), anxiety recognition using physiological signals (doi:10.3390/electronics8091039) and stress recognition at workplaces (10.1007/978-3-319-62404-4_19). Instead of raw signals presented in Figure 1, present a summary of their statistical characteristics, such as amplitude, frequency, energy, etc. I also would suggest to evaluate the reliability of results using Test-Retest Reliability.
One limitation of the study is that the study only includes students from a specific scholarship program at one university, which may not be representative of all university students. Furthermore, it is not reported if the program was adapted for the target population or if it was a pre-validated program, and if it is the case the authors could provide more information about it. In conclusion, the study presents a multicomponent program that shows promise in improving the academic stress management and RSI of university students. However, further research with larger and more diverse samples is needed to confirm these findings and to understand the generalizability of the program.
Reviewer 2 Report
This paper described an interesting research work on evaluating the academic stress of students with a Resilience to Stress Index (RSI). The physiological data were collected from student participants during an eight-week program that contains three phases. The following comments may be considered for further improvements:
(1) In Section 3.1, 71% women and 29% men participated in the stress evaluation. It is therefore necessary to discuss whether the experiment results were gender-dependent.
(2) It is suggested declaring that the student participants were free of cardiovascular diseases or other mental disorders, because such diseases would mislead the incorrect physiological measurement.
(3) In Section 3.2, it should include the setting details of the psychophysiological stress profile measurement, such as the sampling rate, digitalization resolution, amplification, preprocessing settings (baseline filtering, artifact removal).
(4) Figure 1 caption should provide the subject gender and age information.
(5) In Page, Line 319, the reference citation is missing.
(6) Please clarify why only two principal components of PCA were used? Is there any criterion to determine the best appropriate principal components.
Round 2
Reviewer 1 Report
The authors have improved the manuscript, but more changes are still required to improve the quality and scientific soundness. Here are my further concerns:
1. The limitations of the study must be extended and discussed in more detail. There is a high gender inbalance in the participants, which may have influenced the results.
2. The statistical analysis must be improved. The authors found statistically significant differences in RSI values when comparing pre-test and post-test. Can the authors confirm it in both groups (women and men) when considered separately? Statistical testing such as ANOVA can be used.
3. The words "male"/"man" and "female"/"woman" should be phased out in science because they reinforce ideas that sex is binary. For the sake of inclusivity, the terms such as "sperm-producing individual" or "egg producing individual" should be used to avoid emphasising hetero-normative views. The authors are suggested to consult https://www.eeblanguageproject.com/repository
4. Figure 7 is not informative. Consider replacing with boxplot.
5. Provide more informative explanation of PCA results. I do not agree that "a mean explained variance of 0.6384 with two principal components is enough to understand the data". At least 0.9 of explained variance is required for satisfactory explainability.
Reviewer 2 Report
The manuscript has been improved and revised with review comments.
Author Response
We would like to thank you for the detailed comments and suggestions to enhance the manuscript.